# Analyzing the Potential Biological Determinants of Autism Spectrum Disorder: From Neuroinflammation to the Kynurenine Pathway

**DOI:** 10.3390/brainsci10090631

**Published:** 2020-09-11

**Authors:** Rosa Savino, Marco Carotenuto, Anna Nunzia Polito, Sofia Di Noia, Marzia Albenzio, Alessia Scarinci, Antonio Ambrosi, Francesco Sessa, Nicola Tartaglia, Giovanni Messina

**Affiliations:** 1Department of Woman and Child, Neuropsychiatry for Child and Adolescent Unit, General Hospital “Riuniti” of Foggia, 71122 Foggia, Italy; savino.rosa@ospedaliriunitifoggia.it (R.S.); sdinoia@ospedaliriunitifoggia.it (S.D.N.); 2Department of Mental Health, Physical and Preventive Medicine, Clinic of Child and Adolescent Neuropsychiatry, Università degli Studi della Campania “Luigi Vanvitelli”, 81100 Caserta, Italy; marco.carotenuto@unicampania.it; 3Department of the Sciences of Agriculture, Food and Environment, University of Foggia, Via Napoli 25, 71100 Foggia, Italy; marzia.albenzio@unifg.it; 4Department of Education Sciences, Psychology and Communication, University of Bari, 70121 Bari, Italy; alessia.scarinci@uniba.it; 5Department of Medical and Surgical Sciences, University of Foggia, Viale Pinto, 71122 Foggia, Italy; antonio.ambrosi@unifg.it (A.A.); nicola.tartaglia@unifg.it (N.T.); 6Department of Clinical and Experimental Medicine, University of Foggia, 71122 Foggia, Italy; francesco.sessa@unifg.it (F.S.); giovanni.messina@unifg.it (G.M.)

**Keywords:** autism spectrum disorder, neuroinflammation, Kynurenine pathway, microglia, oxidative stress, mitochondrial disorder, immune deregulation, QUIN (quinolinic acid), KYNA (kynurenic acid), tryptophan catabolites

## Abstract

Autism Spectrum Disorder (ASD) etiopathogenesis is still unclear and no effective preventive and treatment measures have been identified. Research has focused on the potential role of neuroinflammation and the Kynurenine pathway; here we review the nature of these interactions. Pre-natal or neonatal infections would induce microglial activation, with secondary consequences on behavior, cognition and neurotransmitter networks. Peripherally, higher levels of pro-inflammatory cytokines and anti-brain antibodies have been identified. Increased frequency of autoimmune diseases, allergies, and recurring infections have been demonstrated both in autistic patients and in their relatives. Genetic studies have also identified some important polymorphisms in chromosome loci related to the human leukocyte antigen (HLA) system. The persistence of immune-inflammatory deregulation would lead to mitochondrial dysfunction and oxidative stress, creating a self-sustaining cytotoxic loop. Chronic inflammation activates the Kynurenine pathway with an increase in neurotoxic metabolites and excitotoxicity, causing long-term changes in the glutamatergic system, trophic support and synaptic function. Furthermore, overactivation of the Kynurenine branch induces depletion of melatonin and serotonin, worsening ASD symptoms. Thus, in genetically predisposed subjects, aberrant neurodevelopment may derive from a complex interplay between inflammatory processes, mitochondrial dysfunction, oxidative stress and Kynurenine pathway overexpression. To validate this hypothesis a new translational research approach is necessary.

## 1. Introduction

Autism Spectrum Disorder (ASD) represents a highly heterogeneous neurodevelopmental disease starting during early childhood and characterized by social and communication deficits, stereotypic and rigid patterns of behavior, restricted interests, and impaired sensory integration processing [1]. The prevalence of ASD has dramatically increased during the last two decades from 2–5/10,000 to 1:59 children [2] with high social and economic impact.

Although ASD etiology has not yet been defined, significant genetic, epigenetic and environmental determinants have been identified.

In particular, genetic analyses have revealed not only a strong genetic basis for ASD, but have also highlighted the complexity of such disorders that cannot be simplistically explained by the paradigm “one mutation-one disease”. Moreover, numerous genome-wide studies have suggested multigene interactions and/or rare as well as noncoding mutations as main players in the autistic thaumatological scenario [3].

In parallel, analyses of the genotype to phenotype relationships in ASD [4], identification of the sequences that co-localize with cell-type-specific regulatory regions [5] and the application of bioinformatics tools [6] have paved the way to proteomic analyses in autism. Today, numerous autism-related proteins have been identified and are the object of intensive research efforts in order to understand the phenotypic risks for autism. Examples are MeCP2 protein (methyl CpG binding protein 2) [7], reelin protein [8], and CNTNAP2/CASPR2 (Contactin-associated protein-like 2 protein) [9,10].

Immune activation and prenatal exposure to toxins such as thalidomide and valproic acid have been considered environmental factors contributing to idiopathic ASD [11]. Advanced parental age, low birthweight, preterm delivery, and low Apgar scores were also reported to be the few factors more consistently associated with autism [12].

Clinically, altered neurodevelopment during the first and second trimesters of prenatal life is believed to be an underlying neuropathological cause of ASD. Post-mortem studies have unveiled neuroanatomic and cytoarchitectonic aberrations in various brain regions, including cerebellum, hippocampus, inferior olivary complex, amygdala, entorhinal cortex, fusiform gyrus, and anterior and posterior cingulate cortex, with increased growth of the frontal lobes, thinner cortical minicolumns, and increased dendritic spine density [13]. These aberrations appear to be related to alterations occurring during early pregnancy, such as reduced programmed cell death and/or increased cell proliferation, altered cell migration, abnormal cell differentiation with reduced neuronal body size, abnormal neurite sprouting, and pruning, that cause atypical wiring into the brain. In addition, because neurodevelopmental processes are still active into late prenatal and postnatal life, aberrations involve reduced synapse formation and delayed myelination [14]. The observed abnormal neuronal wiring was previously thought to be characterized by long-range hypo-connectivity and local hyper-connectivity. Recent studies have, instead, shown that abnormal neuronal wiring is characterized by a highly individualized combination of hyper- and hypo-connectivity specific to each ASD patient [15,16].

Despite years of studies, the etiopathogenesis still remains unclear. Consequently, no effective preventive and treatment measures have been identified. In the last decade, an increasing body of research has focused on the potential role of inflammation, oxidative and nitrosative stress, mitochondrial dysfunction and dysregulation of the tryptophan catabolite (TRYCATs) pathway in the onset of psychiatric disorders [17,18,19]. Although the association between inflammation, oxidative stress and mitochondrial dysfunction has been repeatedly demonstrated, understanding the primary mechanism and the predominant pathway associated with specific psychiatric symptoms remains uncertain.

Here we undertake a clinical review of the recent literature on these topics, to investigate and clarify the role of biological pathways in the etiology of autism. We aim to provide a unifying hypothesis that reflects the complexity of brain functioning, to help develop an earlier diagnosis and more specific therapeutic targets.

## 2. Neuroinflammation

### 2.1. Biological Background: The Role of Microglia in the CNS

Experimental evidence has highlighted the mutual interaction between the immune system (IS) and the central nervous system (CNS) activity since the intrauterine neurodevelopmental stages. CNS activity may impact immunological functioning through catecholamines, glucocorticoids, and neurotransmitters. Conversely, pro-inflammatory cytokines, monocytes, macrophages, and T or B lymphocytes from the IS act on CNS too [20]. Numerous components of the innate immune system—including physical barriers, cellular processes such as phagocytosis, humoral components such as complement proteins, macrophages, granulocytes, and natural killer (NK) cells—are involved in the mutual interaction between IS and CNS. Concomitantly, adaptive immunity characterized by specificity, immunological memory, and self/non-self-recognition can act via antigen-specific cells that are B cells and T lymphocytes.

On the whole, in the brain an inflammatory cascade may be generated and alter the physiological functioning of resident cells such as microglia, astrocytes, and neurons [21].

In this context, alterations of microglia physiological activity are of particular relevance to ASDs. Indeed, microglia play essential roles in neurodevelopment and synapse functioning [22,23]. Specifically, during CNS development, microglia produce neurotrophic factors, phagocyte redundant neurons and connections, remove cell debris and control stem cell proliferation, in this way regulating synaptogenesis and neuronal pruning [24]. Disruption in such essential functions at any time from pregnancy to the early postnatal period may lead to neurodevelopmental disorders such as ASD [25]. Post-mortem studies have shown a hyperproduction of microglial cells and increase in their density in ASD subjects, particularly in the dorsolateral prefrontal cortex [26], cerebellum, midfrontal and cingulate gyrus [27], front-insular and visual cortex [28]. These findings have been confirmed by positron emission tomography (PET) studies that detected a stronger signal in cerebellum, midbrain, pons, and fusiform gyrus in ASD subjects with respect to controls [29]. Moreover, autistic brains showed increased short-distance microglia-neuron interaction. While neuron-neuron clustering increased with advancing age, microglia-microglia organization was normal through all ages, suggesting that the aberrantly close microglia-neuron association in autism is not a result of altered microglial distribution but a neuron-specific reaction [25].

Finally, a human transcriptome analysis in control and autistic cortical brains revealed a robust, inverse correlation between two differentially co-expressed genetic clusters, activated microglia genes and synaptic transmission genes [25]. The activated microglia genes were altered in the autistic brains, potentially induced by IFN I, such as IFN-α and IFN-β. This process could impact negatively on progenitor cell proliferation and connectivity, with subsequently altered neural activity and function during postnatal development [30].

However, although microglia involvement in ASD has been widely demonstrated, the correlation between their region-specific activation and autistic symptoms is yet to be resolved. Tetreault found higher microglia density in the front-insular cortex and the visual cortex of autistic autopsy brains, concluding that this increase in microglia is likely to be present throughout the brain [28]. In the amygdala, which is best-known for its role in emotional learning [31], increased microglial activation and greater number of oligodendrocytes were observed in a few cases of ASD post-mortem brains [32]. In an animal model of ASD, with duplication of human chromosome 15q11-q13, a microglial alteration in the basolateral amygdala during early development has been correlated to anxiety-related behavior in adolescent mice [33]. Instead, in the prefrontal cortex, which is involved in cognitive control and subsequent behavioral outcomes [31], an altered microglia spatial organization around neurons, as well as a markedly microglia activation have been found [26]. Unfortunately, these studies have not been replicated and need to be interpreted with caution.

### 2.2. Early Infections as Triggers for Immune Deregulation

Epidemiological studies have identified prenatal immune activation linked to infections as a significant risk factor for psychiatric disorders [34]. Moreover, early-life infections are considered as trigger factors for immune alterations acting under certain genetic conditions. During pregnancy, the maternal immune system cannot protect the developing fetal brain efficiently because of the physiological fetus’ immunosuppression [35].

In this scenario, psychiatric disorders may arise from the indirect effect of microglia activation, CD4+ and CD8+ T Cells, and the consequent proinflammatory cytokine production that also plays a role in brain development and synaptogenesis [5,36]. Aberrant levels of proinflammatory cytokines such as interleukin 6 (IL-6), TNF-α and monocyte chemotactic protein-1 (MCP-1) were found in brain specimens and cerebrospinal fluid (CSF) [27], as well as in amniotic fluid [37] of children and adults with ASD.

In animal models, an excess of cytokines seems to “sensitize” the immune system, according to the theory of immune system kindling and sensitization as a theoretical basis for a stress-induced inflammatory response in psychiatric disorders proposed by Muller et al. in 2015 [38]. According to this theory, after the primary immune response to specific stimuli such as stress or infection, successive re-exposure can cause a higher release of cytokines or can activate the same immune process also due to a weaker stimulus. Consequently, in individuals sensitized during pregnancy a re-infection or a re-activation of silent infections would induce a massive release of proinflammatory cytokines with secondary neurotransmitter activation disturbance [38].

### 2.3. Autoimmunity and Genetics

Autoimmunity and genetic susceptibility can activate and sustain a chronic subclinical inflammatory condition [39]. On the other hand, an increased frequency of allergy and autoimmune disorders has been found among mothers of ASD children with different effects on clinical severity of social deficit and behavioral alterations. Indeed, it has been shown [40,41] that in families with multiple autoimmune disorders, the odds ratio for a risk of a child with ASD increases. The autoimmune prevalence was higher in mothers and first-degree family members of ASD subjects, with type I diabetes (T1DM), rheumatoid arthritis (RA), hypothyroid and systemic lupus erythematosus (SLE) being the most common disorders found [40]. Maternal diagnosis of celiac disease, psoriasis (especially in the four years surrounding pregnancy), or allergy/asthma also conferred an increased risk of ASD diagnosis in offspring, as well as greater clinical impairment [41,42].

A strong correlation was found between family recurrence of Hashimoto’s thyroiditis (especially in mothers) and ASD offspring [41,43]. A large case-control study in Finland identified a significantly increased risk of developing ASD in children who were born to mothers positive for anti-thyroid peroxidase antibodies (TPO-Ab+) during pregnancy [44]. A 2016 meta-analysis of mainly case-controlled studies confirmed significant positive associations of ASD with maternal autoimmune thyroid conditions [45]. Familial T1DM and autoimmune thyroid disease were also associated with higher rates of regressive autism versus those with developmental delays evident during infancy [41].

Additional immune alterations have been described in ASD individuals including imbalances in antibody levels [46,47], higher neural autoantibodies and pro-inflammatory levels [36], abnormalities in the ratio of Th1/Th2/Th17 cells [48], and reduced natural killer (NK) cell activity [49].

Genetically, immune dysfunction in ASD has been suggested to be linked also to the MHC genes for class I, II, and III molecules, considering that the class I and II molecules are involved in antigen presentation and the development, refinement, maintenance, and plasticity of the brain [50,51]. Different human leukocyte antigen (HLA) haplotypes such as HLA DRB1 [52], HLA DR4 [53], and HLA DR11, seem to be more frequently associated with ASD, although not yet identified as disease markers. In particular, the HLA-DRB1 *11-DQB1*07 haplotype was more prevalent in ASD patients, possibly linked to gastro-intestinal-inflammatory alterations given that this haplotype associates with pediatric celiac disorders [54]. A recent HLA genotyping study showed, in an ASD cohort, a significative expression of HLA-Cw7 [55], interestingly linked to a higher incidence of allergies, food intolerances, and chronic sinusitis. These findings seem to be significant since HLA-C is known to be the ligand of killer-cell immunoglobulin-like receptors (KIRs). KIRS are a family of cell surface proteins situated on natural killer cells, which regulate their activation or inhibition to guarantee self-tolerance [56,57]. Torres et al. found in an ASD population an over-expression of haplotype KIR 2DS1/HLA-C2 corresponding to abnormal NK cells activation [56,57]. Within the HLA class III region, there is a complement C4B null allele, which confers a relative risk for the development of ASD [56].

Functional polymorphisms of macrophage inhibitory factor (MIF), which influences innate and adaptive immune responses, have also been related to ASD [46]. Increased sera concentrations of MIF correlated with worsening behavioral assessments in individuals with ASD [46].

In addition, genetic polymorphisms in immune-related gene loci, including MET (met proto-oncogene) receptor tyrosine kinase gene, threonine kinase C gene PRKCB1, CD99 molecule-like 2 region (CD99L2) [58], Jumanji AT rich interactive domain 2 (JARID2) [58] gene, the thyroid peroxidase gene (TPO) [58], tuberous sclerosis proteins 1 and 2 (TSC1-TSC2) genes [59], and phosphatase and tensin homolog (PTEN) [41], have been associated with ASD. In this regard, TSC1, TSC2, and PTEN mutations are of special importance since they might underlie the above mentioned decreased apoptosis with consequent increase of cell proliferation that characterizes autistic brains. Indeed, TSC1, TSC2, and PTEN negatively regulate cell proliferation and promote cell death [60,61]. It is also of note to observe that the complex TSC1-TSC2 exerts its physiological action by negatively regulating Mammalian Target of Rapamycin (mTOR) signaling [62], and that the ultimate target of the mTOR signaling pathway is the ribosome [63]. Interestingly, the copy number of ribosomal genes was hypothesized as a factor of autism development and severity [63]. Hence, these data add copy number variants to the single-nucleotide variants causing loss-of-function or missense changes in the mosaic of gene alterations implicated in ASDs [64,65].

Moreover, some studies evidenced innate immunity impairment as ASD promoter. Enstrom et al., demonstrated an improved responsiveness to signaling via select TLRs (TLR 2, TLR 4) and conversely a decreased production of cytokines following stimulation of TLR 9 [49]. Gene networks involved in immune processes seem to be overexpressed in the ASD brain [66,67], linked to atypical expression of nuclear factor kappa-light-chain-enhancer of activated B cells (NF-κB) in a number of cell types in ASD including neurons, astrocytes, and microglia [68,69].

### 2.4. Inflammation, Mitochondria and Oxidative Stress

Oxidative stress, mitochondrial dysfunction and immune deregulation in ASD subjects was well established [70,71,72] in mutual interactions. The interaction between redox homeostasis, inflammation and mitochondrial function may be important in disease initiation, progression and treatment. Reactive oxygen species (ROS) and reactive nitrogen species (RNS) produced by the innate immune system and microglia as defense mechanism against exogenous factors activate the release of pro-inflammatory cytokines such as IL-6 and TNF-alfa [73].

Specifically, oxidative stress activates different transcriptional factors such as NF-KB, increasing the expression of pro-inflammatory cytokines [72] with massive release of oxygen and nitrogen reactive molecules, perpetuating the so called “Auto-toxic Loop”. More specifically, between chronic oxidative stress and systemic inflammation “a self-sustaining and self-amplifying relationship” is established able to amplify the existing neuroinflammation and to catalyze the progression of pre-existing pathology [74]. Furthermore, the oxidative and nitrosative damage to DNA, proteins and lipids produces new molecular patterns acting as new antigens, enhancing once again this vicious circle [74], suggesting the cause for the increased circulating autoantibodies against the neuronal and glia filaments reported in ASD [75] and explaining the reason for the lower concentrations of antioxidants in autistic patients [76].

Specifically, in 2012, Frustaci et al. [76] in a meta-analysis from 29 studies of blood samples from ASD subjects reported reduced levels of antioxidant such as glutathione (GSH), glutathione peroxidase, methionine, and cysteine associated with increased levels of oxidized glutathione. Furthermore, Goldani [77] found decreased levels of transferrin and ceruloplasmin in patients with ASD.

Negative correlations between ASD severity and plasma GSH, catalase and superoxide dismutase (SOD) as well as serum Nicotinamide Adenine Nucleotide (NAD)+ and ATP levels are also evident, suggesting redox dysregulation in association with mitochondrial dysfunction. A decrease in total antioxidant capacity is also found in adolescents with Asperger’s syndrome [78] indicating that redox alterations are relevant across the spectrum of severity of ASD-like symptoms [39,76,79].

A recent systematic review of the role of mitochondria in ASD found that all included studies showed disruption of the electron transport chain [80]. The prevalence of formal mitochondrial disorder in ASD is around 5%, about 500 times higher than in the general population [72]. The prevalence of abnormal mitochondrial dysfunction biomarkers is much higher than formal mitochondrial disorder in ASD, being estimated to be evident in about 80% of ASD diagnoses. Mitochondrial dysfunction will crucially contribute to, and be modulated by, redox dysregulation [74,81].

Mitochondrial dysfunction can be classified as either primary or secondary. Primary mitochondrial dysfunction is defined as a congenital mutation of nuclear DNA or mtDNA, with consequent alteration of energetic systems, necessary for ATP production [72,79]. Secondary mitochondrial dysfunction, instead, refers to other metabolic or genetic abnormalities that, secondarily, lead to mitochondrial impairment. When formal mitochondrial disorder associates with ASD, there is a significantly increased prevalence of developmental regression, seizures, motor delay, gastrointestinal abnormalities, female gender, and elevated lactate and pyruvate versus the general ASD population [79]. In this case mitochondrial dysfunction is suggested to connect the diverse range of symptoms associated with ASD, partly driven by oxidative stress and decreased anti-oxidants [79].

The fact that only 23% of children with ASD/mitochondrial disease have a known mtDNA abnormality suggests that mitochondrial dysfunction may be acquired [72,79]. Mitochondrial function can be potentially inhibited by inflammation mediators, with consequent long-term impairment. In an attempt to explain the role of mitochondrial dysfunction and dysreactive immunity in neurodevelopment disorders, Palmieri and Persico [82] proposed a pathogenetic model for ASD. According to their hypothesis, an excess of inflammatory cytokines, such as TNF-α and IL6, and cytokine receptors, such as CD38, would activate a flux of calcium second gradient towards cytoplasm. Increased cytoplasmic Ca2+ exceed the mitochondrial capability to balance calcium homeostasis, and leads to persistent mitochondrial damage, with consequent oxidative stress and abnormal oxidative phosphorylation. Conversely, an increase in intracellular Ca2+ and mitochondrial dysfunction would in turn be able, possibly through NO and/or oxidative stress, to exacerbate an immune response in the psychiatric patient’s brain [82].

Reduced energy availability negatively affects neurodevelopment, in terms of neuronal connectivity, neurotransmission, myelination, and neuronal differentiation [79,83]. Much evidence supports the role of mitochondria as mediators of brain-derived neurotrophic factor (BDNF) action on synaptic plasticity. BDNF promotes synaptic plasticity [72,84] because it is able to enhance mitochondrial energy production [85] by implementing the activity of respiratory complex I [83]. Interestingly, altered levels of BDNF, have been found in serum and in the brain of patients with neurodevelopmental disorders [86]. Mitochondrial dysfunction would also impair synaptic neurotransmitter release, thus neurons with high firing rates, such as (Gamma-Aminobutyric acid) GABAergic interneurons, or glutamatergic neurons, result in being the most adversely affected. Consistent with this, excitatory and inhibitory imbalances have been found in ASD [87]. Mitochondria regulate glutamatergic system, possibly affecting N-methyl-D-aspartate (NMDA) signaling, whose impairment has been correlated to altered synaptic plasticity and cortical micro-circuitry [88].

Furthermore, mitochondria play a role in calcium homeostasis [89]. Mitochondria actively transport calcium from cytoplasm to the matrix, regulating its release and participating in different calcium mediated signaling processes. Then the main second messenger calcium ions directly and indirectly regulate the short- and long-term neuronal plasticity and neurotransmission [89]. It is also essential in mediating neuronal apoptosis, which occurs during development and adult cell turnover. Indeed, the cytoarchitectonic abnormalities found in autistic brains have been correlated to decreased apoptosis with consequent increase of cell proliferation, abnormal cell migration, and altered cell differentiation with reduced neuronal size during the first semester of pregnancy [13,90,91].

## 3. The Kynurenines Pathway

In recent years new etiologic putative factors for neurodevelopmental disorders have been identified in the kynurenine pathway (KP), the major route for the catabolism of tryptophan (TRP) involved in neuroprotection and immunomodulation via the “kynurenines” [92,93]. TRP is the main source of cerebral serotonin and melatonin, as well as of de novo NAD. TRP metabolism is considered the “heart of the whole complex network between the immune and neuroendocrine systems” [94] since it has been implicated in many physiopathology processes, such as neurotransmission, inflammation, oxidative stress and the immune response [95]. Recent reports identified in microbiota alteration a significant causative role for neurodevelopmental disorders, considering that TRP is the main neuromodulator between gut and brain [96,97].

### 3.1. Tryptophan-Kynurenines Metabolism

TRP can be metabolized along two pathways involving methoxy-indoles or kynurenines (KYNs). The KYN pathway is the major degradative metabolic route for TRP and a key regulator of the immune response [39]. Metabolism of TRP along the methoxy-indole route results in the formation of 5-hydroxytryptamine (5-HT/serotonin) and 5-methoxy-N-acetyltryptamine (melatonin) [18,75].

The first and rate-limiting step into the KYN metabolism is regulated by indole-2,3-dioxygenase (IDO) and, to a lesser extent in the brain, by tryptophan-2,3-dioxygenase (TDO) which convert tryptophan to N-formyl kynurenine [98] (Figure 1).

TDO is predominantly expressed in liver and kidney tissues, while IDO is diffused all over the human body’s tissues. N-formyl kynurenine is later metabolized to L-kynurenine (L-KYN) by kynurenine formamidase. Kynurenine (KYN), a pivotal metabolite in the pathway, can, in turn, be metabolized via two distinct pathways: the neuroprotective kynurenic acid (KYNA) branch via the enzyme KYN amino transferase (KAT), and the neurotoxic branch leading to the production of 3-hydroxy-L-KYN (3-HK) and quinolinic acid (QUIN) via KYN monooxygenase (KMO) [99]. QUIN is eventually catabolized to NAD by the action of quinolinic acid phosphoribosyl transferase (QPRT) [93,95] Figure 1.

### 3.2. Neuroactivity of Kynurenines

Why should KP be considered as a potential key tool in the complex etiology of autistic disorders? The answer could be found in the wide neuroactivity of KP enzymes, metabolites and final catabolites (TRYCATs) involved in the main biological pathways underpinning psychiatric conditions.

For decades KYNA has been considered to have a neuroprotective role since it acts as an antagonist of the N-methyl-D-aspartate (NMDA) [100].

As a NMDA antagonist, KYNA may serve as an “endogenous anti-excitotoxic” agent [93]. On the other hand, KYNA acts as neuro-mediator in physiological concentrations, which when elevated can inhibit NMDA receptor hyperactivation and ameliorate glutamate excitotoxicity in pathological conditions [99,101].

Additional sites of KYNA agonist activity relating to inflammation have been reported, such as the G-protein coupled receptor (GPR35), and the aryl hydrocarbon receptor (AhR) [102]. In particular, GPR35 modulates cAMP production and inhibits the N-type Ca channels of sympathetic neurons and astrocytes, suppressing several inflammatory pathways [103]. AhR is a xenobiotic-sensing receptor that promotes the metabolism of environmental toxins and appears to play an important role in terminating cytokine release in several cell types including macrophages [61].

KYNA is also considered a potential endogenous antioxidant, since it has been shown to scavenge different free-radicals, which contribute to create oxidative stress and consequent mitochondrial dysfunction [93]. Not surprisingly, the KYNA concentration is high during fetal development, decreasing in the immediate postnatal period and remains stably low in adulthood. It has been proposed that a large amount of KYNA could be neuroprotective during gestation or delivery while a rapid postnatal decrease is necessary to disinhibit NMDA receptor function allowing these receptors to lead the normal brain development [100].

Among KP metabolites, QUIN is likely to be one of the most important in terms of biological activity and toxicity. Many studies have shown that QUIN is involved in the neurotoxicity during several inflammatory brain diseases [25]. However, QUIN is a neurotoxic KP metabolite since it activates NMDA receptors (NMDAr), increases neuronal activity, and elevates intracellular calcium concentration [104,105]. This leads to consequent impairment of cytoskeleton homeostasis, with decrease of mitochondrial function and eventually cell death induction [95]. QUIN causes selective neuronal lesions into the hippocampus and striatum because it acts selectively on NMDA receptor subtypes containing the NR2A and NR2B subunits [106], resulting in neurochemical, behavioral, and pathological changes [104].

High cerebral QUIN levels could alter the excitation/inhibition ratio of the NMDA receptor. As NMDA agonist, QUIN tends to increase neuronal glutamate release, inhibiting its uptake by astrocytes, and blocking astroglia glutamine synthetase, leading to excessive microenvironmental glutamate concentrations and thus neurotoxicity [107].

On the other hand, during the second year of life autistic symptoms can often emerge while the glutamate activity is high, thus suggesting its involvement in excitotoxic neuronal damage and/or impairing neuronal connectivity [108]. This suggestion is coherent with the current hypothesis according to which children with ASD have “noisy” and unstable cortical networks related to the disproportionate high level of excitation, or disproportionate weak inhibition [108,109]. In this picture, QUIN has been proposed to play a central role in glutamate excitotoxicity, in altered synaptic plasticity, neural network oscillatory abnormalities, seizures and epilepsy, learning and memory impairments, visual system abnormalities, dyspraxia, behavioral alterations, and social dysfunction, all commonly identified in ASD patients [16].

High concentrations of glutamate and glutamine were found in the amygdala-hippocampal complex and auditory cortex of ASD children [110]. Additionally, memantine, an NMDA antagonist, has been shown to have therapeutic benefits on ASD patients thus supporting the potential use of the QUIN antagonist against autistic symptoms [111].

In addition, QUIN can contribute to free radical generation and oxidative stress that are linked to lipid peroxidation activation, to the interaction with Fe2+ to form QUIN-Fe2+ complexes promoting ROS generation [112], to the increase of NOS and the decrease of SOD activity and GSH levels [95].

Both KYNA and QUIN also regulate α7-nicotinic acetylcholine (α7nACh) receptors, acting respectively as antagonist and agonist [113]. Cholinergic neurotransmission has been implicated in regulating neuronal growth and differentiation, and several studies reported a highly regulated expression of α7nACh receptors in the developing brain, during critical periods for synaptic plasticity [114]. In this light, α7nACh receptors underlie alterations in specific neuropsychological skills including attention, learning, memory and motivation [115], commonly impaired in ASD [1]. In the brain, α7nACh receptors are abundantly located at the presynaptic terminals [69,116], and in vivo bi-directional changes in KYNA regulate the release of neurotransmitters, including glutamate, GABA, dopamine and ACh. In this regard, increased levels of KYNA may induce an excessive blockage of α7nACh receptors on cortical interneurons, projecting neurons, and various cortical afferents [116], resulting in neurotransmitter systems dysfunction and cognitive deficits [117].

### 3.3. Kynurenine Pathway Enzymes and Inflammation

In general, inflammatory processes significantly alter the balance between neurotoxicity and physiology by shifting tryptophan metabolism to kynurenine production. IDO is considered the principal enzyme contributing to the production of kynurenine in inflammatory diseases produced by different cell types, including macrophages, microglia, neurons and astrocytes [92,93].

Infections may induce an immune response that increases the production of several proinflammatory cytokines, including TNF-α, IL-6, IL-1β, and IFN-γ. These cytokines activate the first rate-limiting KP enzymes, including IDO1 and tryptophan-2,3-dioxygenase (TDO), as well as the second rate-limiting KP enzyme, KMO [92,118]. In addition, psychological stress increases synthesis and secretion of adrenal cortisol hormones, which affect the KP in the peripheral system (implementing TDO activity) and the brain, shifting tryptophan catabolism towards QUIN [118]. Moreover, the hypothalamus-pituitary-adrenal axis seems to be dysregulated in ASD children [119], with high rate of internalizing symptoms [118,120].

Interestingly, Williams and colleagues demonstrated in a rabbit model that maternal inflammation shunts tryptophan metabolism away from the serotonin to the kynurenine pathway, potentially leading to excitotoxic injury and impaired development of serotonin-mediated thalamocortical fibers in the new-born brain [99].

From this viewpoint, the pro-inflammatory status in autistic disorders would activate the enzymes IDO and KMO in microglia, with consequent dysregulation of KP pathway and secondary depletion of serotonin and melatonin (MLT) [39,75,121] (Figure 2).

### 3.4. Tryptophan, Serotonin and Melatonin

Tryptophan is transported from the blood across the blood-brain barrier (BBB) by a competitive transport carrier that is shared by several large neutral amino acids (LNAAs). Once in the brain, tryptophan is converted into serotonin (5 HT) that represents the principal neurotransmitter for “the social brain” since it regulates social cognition and decision-making. Decreased central serotonin also associates with increased aggression [122] linked to altered amygdala-cortex interactions [75], and its depletion exacerbates ASD symptoms, including stereotypes and repetitive behaviors [123].

During brain development, 5-HT has been shown to modulate numerous events, including cell division, neuronal migration, cell differentiation, and synaptogenesis [124,125]. Serotonin depletion during the neonatal period increased the thickness of the cerebral cortex, commonly found with an increased cerebral volume in autistic patients [126]. Abnormalities in synaptic connection induced by altered intracerebral serotonin concentration greatly reduce the number of dendrites in the sensory cortex and in the hippocampus [126], fundamental areas in regulating motivation, emotion, learning, and memory [126,127].

In humans acute tryptophan depletion inhibits serotonin synthesis [128]. A recent meta-analysis of human tryptophan depletion studies [129] highlighted that reduced tryptophan availability mainly affects attention of episodic memory and memory for verbal information [128,129]. As regards ASD children, dietary tryptophan deprivation was found to exacerbate autistic symptoms [130].

On the contrary, elevated dietary tryptophan has a suppressive effect on aggressive behavior and post-stress plasma cortisol concentrations in vertebrates. Acute stress, in contrast to long-term stress, and some immune challenges may increase the tryptophan concentration. These effects are believed to be mediated by activation of the sympathetic nervous system even though all mechanisms involved are not well understood [131,132].

Moreover, it has been suggested that temporary dynamic or spatial differences in regional cerebral blood flow, for example during hypoxia (a ASD risk factor), may influence the rates of tryptophan uptake within specific regions of the CNS [133,134].

Lower tryptophan and KA levels differentially characterize childhood autism and intellectual disability disorder from Asperger’s syndrome that, instead, is marked by elevated tryptophan and lower 5-HT synthesis [135].

Melatonin (MLT) is nocturnally synthesized in the pinealocytes by N-acetylation and subsequent o-methylation of 5-HT. It is released by the pineal gland into the systemic circulation and readily passes the BBB [39], playing a key role in circadian rhythmicity including sleep pattern regulation. Poor sleep is one of the most common comorbidities in ASD with rates up to 80% [136], probably linked to the high Orexin A plasmatic levels [137].

However, MLT is released by many different cell types, including the enterochromaffin cells of the gut, glia, and immune cells, influencing many other cellular processes. In particular, MLT acts as a powerful anti-inflammatory, antioxidant and optimizer of mitochondrial functioning, and as stabilizer for the gut-barrier [39]. Its neuroprotective and antioxidant effects were proved, respectively, preclinically on rats’ peripheral neural fibers [138], and clinically on ALS (amyotrophic lateral sclerosis) patients [139], in both cases by administering high dose of MLT.

It plays an important role in tryptophan metabolism by affecting both key enzymes of the two pathways. In particular, it has been demonstrated that melatonin treatment induces the expression of indole-2,3-dioxygenase 1 (IDO1) and enhances the activity of the IDO1 promoter while decreasing the expression of arylalkyl amine N-acetyl-transaminase (AANAT), the enzyme responsible for 5-HT catabolism [140].

In addition, higher melatonin concentration was able to stimulate 5-HT release from the pinealocyte in a preclinical study on rats [141].

### 3.5. Kynurenine Profile in Autism Spectrum Disorders

Various degrees of neuroinflammation attributable to different mechanisms have been widely described in ASD. Some investigated the proinflammatory cytokine profile in patients with autism [142,143], while others focused on glutamatergic imbalance and toxicity as neuroinflammation marker [144].

As explained above, QUIN is a glutamatergic excitotoxin which is produced in answer to inflammation as a way to increase NAD+ levels so to provide more energy to the cell [145]. In human neurons quinolinate phosphoribosyl-transferase (QPRT), which metabolizes it leading to NAD+ production, could be saturated by QUIN at high concentrations so as to result in QUIN accumulation [146]. Interestingly NAD+ levels were found to be decreased in ASD children [147].

In 2016, Lim [148] studied immunological and KP profile among ASD patients. He reported, despite no significant changes in TRP concentrations, an elevation in the ratio of Kynurenine to TRP (K/T) indicating KP activation in the patient group compared to heathy controls. More specifically, there was no significant difference in KA levels between the two cohorts, whereas QUIN serum concentration was significantly increased in the ASD patient group. Picolinic Acid (PA), a neuroprotective mediator, was also found to be decreased. These findings demonstrated for the first time an alteration in KP metabolism in ASD [148]. In agreement with this, the urinary metabolomic analysis of Italian ASD patients revealed a significant increase of xanthurenic acid, QUIN, in parallel with a decrease in kynurenine and kynurenic acid [149]. These considerations mean that events largely limited to the periphery but which, via IDO or TDO, alter tryptophan/kynurenine ratios in the blood, can produce significant secondary changes in the amounts of kynurenine metabolites in the CNS contributing no doubt to the effects of immune activity and stress as noted above [150].

Genetic analyses revealed tryptophan 2,3-dioxygenase polymorphism in autism [151]. Likewise, tryptophan 5-hydroxylase (THP) gene polymorphism has been clearly associated with autism and other neurological disorders [152], Cascio et al. [153] detected, in the ASD population, pathogenic variants in the genes encoding the heavy (SLC3A2) and light subunits (SLC7A5 and SLC7A8) of the large amino acid transporters (LAT) 1 and 2. LAT1 and 2 are responsible for the transportation of tryptophan and other large aromatic amino acids across the blood–brain barrier and are expressed both in blood and brain. Such abnormalities could affect tryptophan availability during brain development, indirectly altering serotonin and KP activity. Furthermore, the tryptophan metabolism-related enzyme quinolinate phosphoribosyl-transferase (QPRT) affects the regulation of genes and gene networks previously implicated in ASDs [154] and, at the same time, a deletion in 16p11.2, the chromosomal regions responsible for QPRT codification, has strongly been associated to ASD [154].

Of no less importance, ASD-associated genetic variations are found in genes coding for NMDAr subunits [155], on which kynurenine neuroactivity is carried out.

This evidence suggests a link between neuro-inflammation, ASD and kynurenines. Since kynurenines may act directly and indirectly on different neurotransmitter systems, and in turn, are influenced by some environmental conditions such as infection, stress, and inflammation, it can be assumed that kynurenines may play a key role in ASD etiology [102] (Figure 3).

### 3.6. Future Perspectives and Drugs

Since KP interferes with many biological processes, it could be considered a crossroad in ASD etiology. Targeting KP with new specific agents may prevent disease or secondary conditions through appropriate pharmacological manipulations. In this scenario, knowledge of the kynurenine pathway will be central to avert ASD rather than treating its symptoms.

Many drugs could interfere with crucial stages of KP. Neuroactive molecules such as ketamine, phencyclidine, memantine, and galantamine, acting on glutamatergic and cholinergic transmissions, were differently tested to improve ASD symptoms with cautious, but still promising, results.

Indeed, it is already possible to manipulate TRP metabolism, even directly, since KP enzymes are already druggable. Three classes of inhibitors are currently in various stages of development respectively targeting KATII, KMO and IDO.

A KAT II inhibitor can already be administered orally and crosses the BBB, substantively counteracting KYNA levels in rats models [156]. Their usage is under investigation because of the supposed antipsychotic effect of reducing KYNA levels to contrast with burst-firing NMDA-induced dopamine in the forebrain. Since ASD and schizophrenia share some anatomic and functional vulnerabilities in prefrontal areas [157], KATII may become of interest from a therapeutic point of view in future years.

On the opposite branch of KP, KMO inhibitors are proposed to increase KYNA levels in animal models where they were showed to have neuroprotective, anticonvulsant and anti-dyskinetic properties in vivo. Unfortunately, they are not able to cross the BBB so that KMO peripheral inhibition leads to a moderate, but persistent, increase in brain KYNA levels, worth further experiments [158].

Because of its role in the development of immune tolerance to tumor antigens, the suppression of T and natural killer cells, and the generation and activation of T regulatory cells, IDO was extensively studied and IDO inhibitors, already found to be effective in preclinical studies are currently employed in clinical trials [159]. Since the immunomodulation line of research also crosses over with neurodevelopmental disorders, we suggest keeping an eye on IDO inhibitors as potential new tools also in ASD.

Lastly, melatonin is universally strongly recommended as first line treatment for sleep disorders in ASD patients [160]. Moreover, its benefits social communication, stereotyped behaviors and adaptation to environmental changes as a result of a “synchronization of internal biological clocks” which has been recently demonstrated [161]. Beyond its symptomatic effects reported above there is evidence of decreased MLT levels in ASD patient, which suggest it may function to replace a deficiency in this case, especially when routine testing for melatonin levels will be available for clinical practice.

As neurodevelopmental disorders occur in the very early stages of life, it is even more important to focus on drug safety when it comes to new therapeutic tools. Among the TRP-related molecules only melatonin has proved to be safe even when administered at a high dose [162]. With regard to its anti-inflammatory, antioxidant and pro-immunity potential as well as its connection role between KP and the 5-HT pathway, we propose MLT as the “new old-fashioned” molecule worthy of being considered from new vantage points (Figure 4).

## 4. Conclusions

The immune system plays an important role in neurodevelopment at multiple neurobiological levels. An aberrant inflammation response, in the very early stage of brain development, determines focal or diffuse neurological damage, and leads to subsequent mental disorder.

We propose ASD to be the result of a complex interplay between genetic susceptibility and environmental predisposition. Our model is based on the hypothesis that, in individuals genetically predisposed and in the presence of specific environmental conditions, a new infection or a reactivation during gestational and/or perinatal age would trigger an abnormal inflammatory response. The inability of the immune system to reduce itself (either because it is immature or because genetically predisposed not to be self-tolerant), induces a cascade of events, which involves, like a domino effect, other biological pathways. The net result is the chronicity of the inflammatory process, through activation of a self-sustaining and self-amplifying “auto-toxic loop” between mitochondrial dysfunction, oxidative stress, and Kynurenine pathway leading, eventually, to an aberrant neurogenesis. As broadly reported, an increased release of pro-inflammatory cytokines mediates:(1)a cascade that “sensitizes” the immune system with subsequent changes in cellular proliferation, activation of microglia and further increases in pro-inflammatory cytokines downstream;(2)increase of ROS and RNS, which leads to oxidative stress and further tissue damage;(3)mitochondrial dysfunction responsible for bioenergetic impairment and consequently neural suffering;(4)activation of Kynurenine pathways with an increase in neurotoxic metabolites and excitotoxicity causing long-term changes in glutamatergic function, trophic support and synaptic function.

These considerations may have therapeutic implications since it is already possible to interfere with TRP metabolism with medication that has already been found to be effective (high dose melatonin, memantine) or by experimenting with KP enzyme inhibitors to lower QUIN levels as new potential symptomatic drugs [142]. Enzymes in the KP are indeed druggable, IDO inhibitors are in various stages of development to treat cancer, and KMO inhibitors have already been found to be effective in preclinical models of neuropathic pain [97].

However, as outlined in the introduction, ASDs refer to a complex and heterogeneous pool of molecules and conditions and it is likely that additional factors play a secondary role in modulating the severity of disorders. Why do individuals develop a specific disease rather than another? Is it possible that the development of a specific psychiatric symptom is associated with a massive involvement of a specific biological pathway?

The major or minor involvement of one or more brain areas is probably responsible for the huge symptom variability observed in affected individuals and it is also conceivable that the role of each pathway in the pathogenesis of mental disorders is potentiated when they act simultaneously.

To better interpret the role of neuroinflammation and Kynurenines in neurodevelopment and to validate the hypothesis previously described, a wide translational research approach is necessary to examine, collectively and simultaneously, epidemiology, genetics, oxidative stress, mitochondrial dysfunction, Kynurenine metabolism and immune deregulation. Such an approach could be able to obtain robust data to clarify underlying pathological processes and identify specific therapeutic targets.

## Figures and Tables

**Figure 1 brainsci-10-00631-f001:**
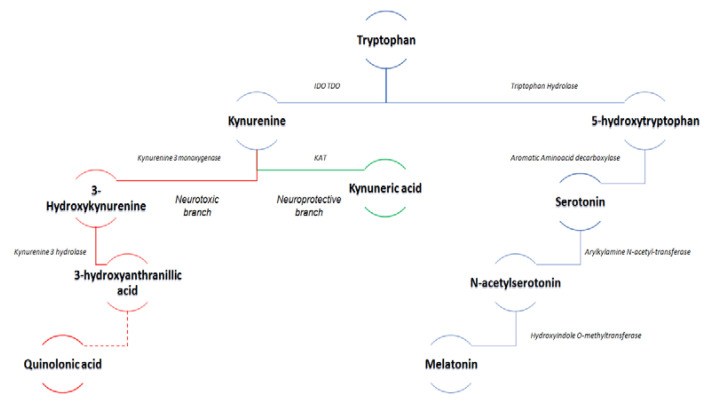
KYNUNERINE PATHWAY: indole-2,3-dioxygenase (IDO) and Tryptophan-2,3-dioxygenase (TDO) lead to Kynurenine synthesis from Tryptophan (TRP), which can be metabolized via two distinct pathways: the neuroprotective kynurenic acid (KYNA) branch via the KYN amino transferase enzyme (KAT), and the neurotoxic branch leading to the production of 3-hydroxy-L-KYN (3-HK) and quinolinic acid (QUIN).

**Figure 2 brainsci-10-00631-f002:**
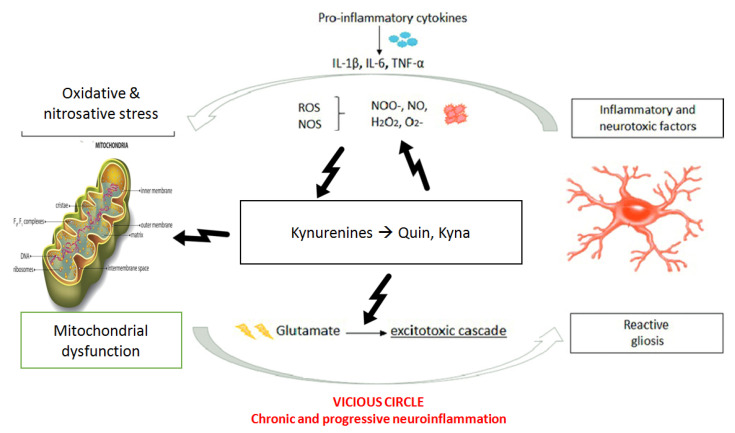
Inflammation significantly shifts tryptophan metabolism to Kynurenine production by activation of activate indole-2,3-dioxygenase (IDO) and Kynurenine monooxygenase (KMO) microglial enzymes. Quinolinic acid (QUIN) is involved in neurotoxicity since it activates N-methyl-D-aspartate (NMDA) receptors, increases neuronal activity, and elevates intracellular calcium concentrations. This leads to the consequent impairment of cytoskeleton homeostasis, decrease of mitochondrial function and finally cell death induction. As an NMDA agonist, it increases neuronal glutamate release, inhibits its uptake by astrocytes, and inhibits astroglial glutamine synthetase leading to excessive microenvironmental glutamate concentrations. In addition, QUIN contributes to free radical generation and oxidative stress.

**Figure 3 brainsci-10-00631-f003:**
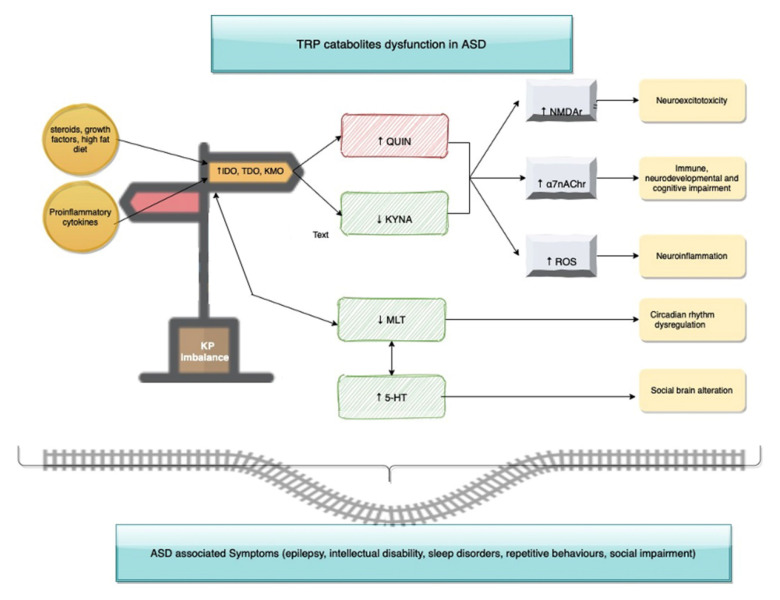
**Kynurenine** (KP) as a crossroad between disrupted routes and pathophysiological conditions that are Autism Spectrum Disorder (ASD) related.

**Figure 4 brainsci-10-00631-f004:**
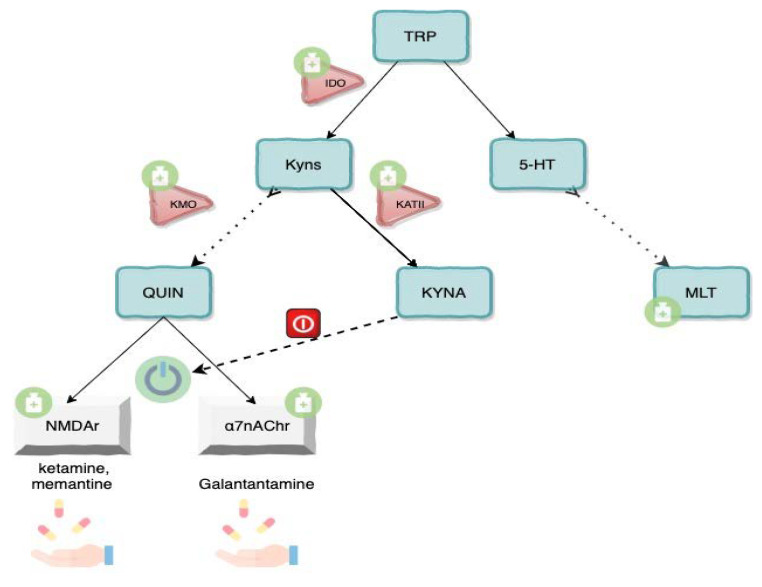
Therapeutic targeting of KP.

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
