# Peer review of "Analyzing the Potential Biological Determinants of Autism Spectrum Disorder: From Neuroinflammation to the Kynurenine Pathway"

_brainsci, 2020, doi:10.3390/brainsci10090631_

Round 1
Reviewer 1 Report
Review of the article
Analyzing the potential biological determinants of Autism Spectrum 2 Disorders: from Neuroinflammation to Kynurenines Pathway 3
Rosa Savino1, Marco Carotenuto2, A. Nunzia Polito1, *, Sofia Di Noia1, Marzia 5 Albenzio6,Alessia Scarinci3, Antonio Ambrosi4, Francesco Sessa5, Nicola Tartaglia4 and 6 Giovanni Messina5
The article is written much better, but I have some comments.
- Line 69: Please, explain the first used abbreviations MeCP2, CASPR 2 and CNTNAP2
- I propose to change the subtitle 2.1 Biological background: the Innate and adaptive immunity in CNS to:
2.1 Biological background: the role of microglia in CNS
Explanation: the chapter deals mostly with the role of microglia in CNS, so it will be more precise. Moreover, under normal physiological (healthy) circumstances, there are no cells or humoral factors of innate or adaptive immunity in CNS. In normal CNS we have only microglia originating from the yolk sack. These are resting cells with self-renewing capacity (previously defined as macrophages) that enter the brain during intrauterine life (8.5 – 9.5 days). They have a great role in CNS development, in maintaining the homeostasis in CNS, in adulthood during neurogenesis, hypothalamic function, importance in memory, and cognition. Also in growth, repair, and metabolism of the tissue. They phagocytose dead cells, apoptotic cells, tissue detritus, … They get into CNS during early intrauterine life, not later, and persist there for all life. The cells and factors of the innate immunity enter from the periphery (eg. Macrophages) to the CNS only under pathological conditions – after some stimulus or disruption of the blood-brain barrier.
- Lines: 107-108: Please change the sentence from: „CNS activity may impact immunological functioning through catecholamines, glucocorticoids, and stimulation of lymphoid organs.“ To: CNS activity may impact immunological functioning through catecholamines, glucocorticoids, and neurotransmitters.“ It is more precise; glucocorticoids do not activate lymphoid organs, they have immunosuppressive activity.
- Lines: 108-109: PLease remove microglial cells and astrocytes from the sentence: „ Conversely, .........“ Explanation: Microglia and astrocyte are cells of CNS, not of IS in the periphery.
- Lines: 112-112: something is missing from this sentence.
- Lines 113-116: Subpopulation of suppressor T cells does not exist more, we have so-called regulatory T cells. Please, shorten the sentence: „ Concomitantly, 113 adaptive immunity characterized by specificity, immunological memory, and self/nonself 114 recognition can act via antigen-specific cells that is B cells and T lymphocytes (namely, 115 cytotoxic, helper, and suppressor T cells).“ To: Concomitantly, 113 adaptive immunity characterized by specificity, immunological memory, and self/nonself 114 recognition can act via antigen-specific cells that are B cells and T lymphocytes.“ It will be enough. And not "is B cell and T lymphocytes, but: are .....
- Line 138: please specify the type of IFN. You wrote only the class of interferon (IFN-I).
- The rest of the article is OK
Conclusion: Although there are some shortcomings in the field of immunology, the article may be published after correction.
Author Response
- Line 69: Please, explain the first used abbreviations MeCP2, CASPR 2 and CNTNAP2
MECP2: methyl CpG binding protein 2
CASPR 2: Contactin-associated protein-like 2
CNTNAP2 Contactin Associated Protein 2 gene
- I propose to change the subtitle 2.1 Biological background: the Innate and adaptive immunity in CNS to:
2.1 Biological background: the role of microglia in CNS
The subtitle has been changed.
3. Lines: 107-108: Please change the sentence from: „CNS activity may impact immunological functioning through catecholamines, glucocorticoids, and stimulation of lymphoid organs.“ To: CNS activity may impact immunological functioning through catecholamines, glucocorticoids, and neurotransmitters.“ It is more precise; glucocorticoids do not activate lymphoid organs, they have immunosuppressive activity.
The sentence has been changed.
4. Lines: 108-109: PLease remove microglial cells and astrocytes from the sentence: „ Conversely, .........“ Explanation: Microglia and astrocyte are cells of CNS, not of IS in the periphery.
Microglial cells and astrocytes have been removed from the sentence.
5. Lines: 112-112: something is missing from this sentence.
Numerous components of the innate immune system - including physical barriers, cellular processes such as phagocytosis, humoral components such as complement proteins, macrophages, granulocytes, natural killer (NK) cells - are involved in the mutual interaction between IS and CNS. Concomitantly, adaptive immunity characterized by specificity, immunological memory, and self/nonself recognition can act via antigen-specific cells that are B cells and T lymphocytes
Numerous components of the innate immune system - including physical barriers, cellular processes such as phagocytosis, humoral components such as complement proteins, m acrophages, granulocytes, natural killer (NK) cells - are involved in the mutual interaction between IS and CNS. 6 .Lines 113-116: Subpopulation of suppressor T cells does not exist more, we have so-called regulatory T cells. Please, shorten the sentence: „ Concomitantly, 113 adaptive immunity characterized by specificity, immunological memory, and self/nonself 114 recognition can act via antigen-specific cells that is B cells and T lymphocytes (namely, 115 cytotoxic, helper, and suppressor T cells).“ To: Concomitantly, 113 adaptive immunity characterized by specificity, immunological memory, and self/nonself 114 recognition can act via antigen-specific cells that are B cells and T lymphocytes.“ It will be enough. And not "is B cell and T lymphocytes, but: are .....
The longer sentence has been replaced
7. Line 138: please specify the type of IFN. You wrote only the class of interferon (IFN-I).
IFN-α and IFN-β
8. The rest of the article is OK
I really would like to thank you for the exhaustive and highly qualified explanation.
Reviewer 2 Report
Thank you for your comprehensive answers. The new title of the manuscript better reflects its content. I have no additional comments.
Author Response
Thank you for your revision
Reviewer 3 Report
In their manuscript, Savino and colleagues present a review of biochemical and immunologic determinants of autism spectrum disorders focusing on metabolic impairments in tryptophan catabolism pathways, especially hydroxyquinaldic acid (kynurenic acid), which is regarded as a universal key to the whole autistic spectrum.
The review contains 156 references and indicates a great amount of job performed by the authors. There are many interesting and actual facts in the review that would be useful for the readers. That is why I would recommend the manuscript for the publication, however, with some minor revisions.
The authors should be more cautious with declaring a universal role for kynurenine pathways in the development of autism and the prospectives of their drug correction. For last two decades, there was a lot of 'magic tablets' against autism announced, but still no effective treatment. Rather, a conclusion should be made that the kynurenine pathways could be potential targets for the majority autistic cases, but being possibly not involved in some other cases.
A common trait in the majority of autistic cases is disturbed protein translational homeostasis in synapses (see Nature 2012;493(7432):371-7 and Nature 2012;493(7432):411-5 – two convincing articles in the same issue of the journal), or 'proteostasis', as more recent reports say. The authors mentioned upregulated cell proliferation in lines 81 and 320, as well as increased growth of the frontal lobes and increased dendritic spine density in lines 78-79, but the underpinning mutations in tuberous sclerosis proteins 1 and 2 (TSC1-TSC2) genes and phosphatase and tensin homolog (PTEN) are mentioned separately, in lines 226-227. This array of evidence should be gathered together and detailed. Interestingly, the ultimate target of the mTOR signalling pathway is the ribosome, and the copy number of ribosomal genes was hypothesized as a factor of autism development and severity (Cells 2019 26;8(10):1151). Other copy number variations (CNV) are proven genetic factors of autism, including those that involve genes affecting serotonin pathways, and a couple of references to articles/reviews about the role of CNV in autism would be appropriate where noncoding mutations are mentioned (line 62). I believe adding all the references mentioned above would significantly improve the manuscript.
The last but not least is language. A native English editor or a professional translator would be recommended. Some ‘footprints’ of Italian language can be seen in phrase structure, word order (for example, ‘Acid Picolinic (PA)’ instead of Picolinic Acid in line 532), grammar, etc.
Author Response
In their manuscript, Savino and colleagues present a review of biochemical and immunologic determinants of autism spectrum disorders focusing on metabolic impairments in tryptophan catabolism pathways, especially hydroxyquinaldic acid (kynurenic acid), which is regarded as a universal key to the whole autistic spectrum.
The review contains 156 references and indicates a great amount of job performed by the authors. There are many interesting and actual facts in the review that would be useful for the readers. That is why I would recommend the manuscript for the publication, however, with some minor revisions.
The authors should be more cautious with declaring a universal role for kynurenine pathways in the development of autism and the prospectives of their drug correction. For last two decades, there was a lot of 'magic tablets' against autism announced, but still no effective treatment. Rather, a conclusion should be made that the kynurenine pathways could be potential targets for the majority autistic cases, but being possibly not involved in some other cases.
Since kynurenines may act directly and indirectly on different neurotransmitter systems, and in turn are influenced by some environmental conditions such as infection, stress, inflammation, it can be assumed that kynunerines may play a key role in ASD etiology
A common trait in the majority of autistic cases is disturbed protein translational homeostasis in synapses (see Nature 2012;493(7432):371-7 and Nature 2012;493(7432):411-5 – two convincing articles in the same issue of the journal), or 'proteostasis', as more recent reports say. The authors mentioned upregulated cell proliferation in lines 81 and 320, as well as increased growth of the frontal lobes and increased dendritic spine density in lines 78-79, but the underpinning mutations in tuberous sclerosis proteins 1 and 2 (TSC1-TSC2) genes and phosphatase and tensin homolog (PTEN) are mentioned separately, in lines 226-227. This array of evidence should be gathered together and detailed.
Interestingly, the ultimate target of the mTOR signalling pathway is the ribosome, and the copy number of ribosomal genes was hypothesized as a factor of autism development and severity (Cells 2019 26;8(10):1151). Other copy number variations (CNV) are proven genetic factors of autism, including those that involve genes affecting serotonin pathways, and a couple of references to articles/reviews about the role of CNV in autism would be appropriate where noncoding mutations are mentioned (line 62). I believe adding all the references mentioned above would significantly improve the manuscript.
I thank the Referee for these insightful suggestions. The data have been gathered into the phrase
” In this regard, TSC1, TSC2, and PTEN mutations are of special importance since might underlie the above mentioned decreased apoptosis with consequent increase of cell proliferation that characterizes autistic brains. Indeed, TSC1, TSC2, and PTEN negatively regulate cell proliferation and promote cell death [60, 61]. Then, it is also of note to observe that the complex TSC1-TSC2 exerts its physiological action by negatively regulating mTORC signalling [62], and that the ultimate target of the mTOR signalling pathway is the ribosome [63]. Interestingly, the copy number of ribosomal genes was hypothesized as a factor of autism development and severity [63]. Hence, these data add copy number variants to the single-nucleotide variants-causing loss-of-function or missense changes in the mosaic of gene alterations implicated in ASDs [64, 65].” (see, please line 227)
The following references have been also added:
(60) Tee AR, Fingar DC, Manning BD, Kwiatkowski DJ, Cantley LC, Blenis J. Tuberous sclerosis complex-1 and -2 gene products function together to inhibit mammalian target of rapamycin (mTOR)-mediated downstream signaling. Proc Natl Acad Sci U S A. 2002;99(21):13571-13576. doi:10.1073/pnas.202476899
(61) Zhao H, Dupont J, Yakar S, Karas M, LeRoith D. PTEN inhibits cell proliferation and induces apoptosis by downregulating cell surface IGF-IR expression in prostate cancer cells. Oncogene. 2004;23(3):786-794. doi:10.1038/sj.onc.1207162
(62) Saxton RA, Sabatini DM. mTOR Signaling in Growth, Metabolism, and Disease. Cell. 2017 ;169(2):361-371. doi:10.1016/j.cell.2017.02.004
(63) Porokhovnik, L. Individual Copy Number of Ribosomal Genes as a Factor of Mental Retardation and Autism Risk and Severity. Cells 2019, 8, 1151.
(64) Velinov M. Genomic Copy Number Variations in the Autism Clinic-Work in Progress. Front Cell Neurosci. 2019;13:57. Published 2019 Feb 19. doi:10.3389/fncel.2019.00057
(65) Dias CM, Walsh CA. Recent Advances in Understanding the Genetic Architecture of Autism [published online ahead of print, 2020 May 12]. Annu Rev Genomics Hum Genet. 2020;10.1146/annurev-genom-121219-082309. doi:10.1146/annurev-genom-121219-082309
The last but not least is language. A native English editor or a professional translator would be recommended. Some ‘footprints’ of Italian language can be seen in phrase structure, word order (for example, ‘Acid Picolinic (PA)’ instead of Picolinic Acid in line 532), grammar, etc.
We have reviewed the English throughout the ms-
This manuscript is a resubmission of an earlier submission. The following is a list of the peer review reports and author responses from that submission.
Round 1
Reviewer 1 Report
Review of the article: Understanding Biological Pathways Behind Autism Spectrum Disorders
Authors: Rosa Savino, Marco Carotenuto, A. Nunzia Polito,*, Sofia Di Noia, Marzia Albenzio, Alessia Scarinci, Antonio Ambrosi, Francesco Sessa, Nicola Tartaglia and Giovanni Messina
Review:
The topic of the article is interesting and actual. However, I have some comments:
Main comments:
It seems as if the different chapters of the work were written by different authors. They are qualitatively different. Some parts are written well, but for example the abstract and introduction, especially the immunological part of the work is written superficially and is full of errors. Even the level of English in these chapters is lower.
- g. Inflammation is a complex biological response designed to defend the host against pathogenic threats, both exogenous (infections) and endogenous (danger signal released by threatened cells). Inflammation is a complex biological response designed to defend the host against pathogenic threats, of both exogenous (infectious) and endogenous (non-infectious) origine.
Comment: Danger signals or „signals of threatening“ or „alarmins“ are produced not only from threatened cells. They are released also from dead cells or after disruption of the homeostasis of the organism.
- The definition of innate immunity is insufficient. In addition to cells, humoral factors such as complement components and others are included.
- TLR do not belong to specific receptors, they are non-specific receptors. They do not recognize a concrete microbe as antigenic receptors on T cells belonging to adaptive immunity (previous term specific immunity), they recognize substances typical for a broad range of bacteria, fungy or viruses (e.g. lipoteich acid, lipopolysaccharide, zymozan, ...). TLR are part of innate immunity (previous term non-specific immunity).
- Terms Cytokines and Chemokines are not capitalized.
- The definition of adaptive immunity is rough.
- Line 78: Microglia, astrocytes and neurons do not belong to innate immunity. And Innate
Immunity should be written as innate immunity.
- The term Peripheral Innate Immunity is not accepted as official terminology
- Can you explain the meaning of sentence: an excess of cytokines seems to “desensitize” the immune system ..... What does mean „desensitized“? The cited publication of Miller et al., (2015) explain the sensitization, not desensitization
- Chapter 2.4 Autoimmunity and Genetics should be reworked. There is very little about genetic predisposition and almost nothing about autoimmunity. Either the chapter should be expanded or omitted.
- The same applies to chapter 2.5 Inflammation, mitochondria and oxidative stress. There is only a mention of mitochondria. Either the chapter should be expanded or omitted.
- Something is missing from the sentence: On the other hand, an increased frequency of allergy and autoimmune 124 disorders among mothers of ASD children with different effects on clinical severity of social deficit 125 and behavioral alterations[27]. 126
Reviewer 2 Report
These authors have a written a manuscript and present hypotheses about the role of kynunerine in ASD. The hypothesis is to build a direct line between this pathway and regulation of neuroinflammation in ASD. While novel, the authors miss opportunities to strengthen their hypotheses which has resulted in a somewhat disjointed paper without clear evidence connecting the various parts of their hypothesis. The reviewer does commend the effort of the authors since this task would be difficult even for native speakers of English. Yes, there are minor language points. Here are the reviewer suggestions for improvement of the manuscript.
1) The intro is much too short and needs more depth about the possible genetic and environmental etiologies. Further the authors need to relate those etiologies to genetic and environmental causes. Do either of these sets of data have any bearig on the KYNA pathways?
2) Does the production of areas of microglial activation in limited regions via post mortem studies suggest its processes may modify only a limited number of ASD symptoms?
3) Some statements line 125-126, 175 are simply not supported by references. The statements are way too strong. For instance, if you maintain TRP is the main signal between gut and brain, you need to back that up with evidence.
4) In general, the KYNA discussion misses the mark. One should relate that to the most likely biologic processes in ASD and certainly present evidence for NMDA receptor involvement. No genetic information is presented.
5) there is quite a a lot of information surrounding the melatonin and serotonin stories in ASD. The current discussion is not sufficient.
6) In general, there is a poor connection between KYNA pathway, regulated biologic processes, and ASD. As an example, what happens when one metabolite such as TRP is up or down during brain development?
Finally the title should be altered and should focus broadly on glutamate or KYNA pathways in the name.
Minor comments: line 73, 86, 106, 145,288 mis-spelled words in English.
Run on sentence: 152-154.
Reviewer 3 Report
In the paper entitled:” Overview: Understanding Biological Pathways Behind Autism Spectrum Disorders”, Savino et al. have summarized the findings of well-established role of neuroinflammation in the pathology of ASD. Moreover, involvement of kynunerine pathway in ASD pathology was investigated, which is a more interesting part of the manuscript. The paper seems to be interesting and is written correctly. However, I recommend to revise the manuscript in several aspects.
Reading the manuscript the question arises what is the main purpose of the work. This should be better emphasized.
The title is inadequate to the data presented.
The chapter no. 2 Neuroinflamation is divided into 5 very short subsection and it is a summary of previous knowledge, unfortunately without novelty. This should be rearranged
- Subsections should be more connected.
- A separate subsection no 2.2. Microglia seems to be completely unnecessary and may be included in the inflammation subsection.
- The 2.5. subsection is titled: Inflammation, mitochondria and oxidative stress and contains only one sentence about mitochondria. This should be corrected.
The part of the manuscript about the potential involvement of the kynunerine pathway in autism spectrum disorders is the most interesting element of the work and should be emphasized.
A few additional figures would be useful in analyzing the data presented, e.g. role of kynunerine pathway in ASD.
Moreover I recommend Authors to add "Chapter 4", which describes future view and perspectives.
Editorial correction is necessary.
